# Evolution of Cardiovascular Risk Factors in Post-COVID Patients

**DOI:** 10.3390/jcm12206538

**Published:** 2023-10-15

**Authors:** Irina Mihaela Abdulan, Veronica Feller, Andra Oancea, Alexandra Maștaleru, Anisia Iuliana Alexa, Robert Negru, Carmen Marinela Cumpăt, Maria Magdalena Leon

**Affiliations:** 1Department of Medical Specialties I, “Grigore T. Popa” University of Medicine and Pharmacy, 700115 Iasi, Romania; irina.abdulan@umfiasi.ro (I.M.A.); robert.negru@umfiasi.ro (R.N.); marinela.cumpat@umfiasi.ro (C.M.C.); maria.leon@umfiasi.ro (M.M.L.); 2Clinical Rehabilitation Hospital, 700661 Iasi, Romania; veronica.feller@gmx.de; 3Department of Surgery II, Discipline of Ophthalmology, “Grigore T. Popa” University of Medicine and Pharmacy, 700115 Iasi, Romania; anisia-iuliana.alexa@umfiasi.ro

**Keywords:** COVID-19, hypertension, obesity, dyslipidemia, long COVID-19

## Abstract

(1) Background: SARS-CoV-2 infection has been a subject of extensive discussion in the medical field, particularly in relation to the risk factors and effective treatment strategies for reducing the negative health outcomes associated with the virus. However, researchers indicate that individuals in the recovery phase after COVID-19 experience a range of symptoms that significantly impact their overall well-being and quality of life. At present, there is insufficient evidence to substantiate the claim that patients in the post-acute phase of COVID-19 are at an elevated risk of developing new-onset hypertension or even metabolic syndrome. The current study aimed to assess the risk of cardiovascular diseases after COVID-19 and the optimal treatment of these conditions. (2) Methods: This research was conducted at the Cardiovascular Rehabilitation Clinic of the Iasi Clinical Rehabilitation Hospital (Romania) between the 1st of September and 31st of December 2022. From a total of 551 patients hospitalized in that period, 70 patients with multiple comorbidities were selected. This study included patients over 18 years old who were diagnosed with COVID-19 within the past 30 days. (3) Results: The included patients were mostly women (62.9%) from the urban area (61.4%). Comparing the post-COVID-19 period to the pre-COVID-19 one, it was observed that the risk of hypertension increased from 69.57% to 90% among the subjects (*p* = 0.005). Risk factors for the new onset of hypertension were identified as age, female gender, and an elevated body mass index. Moreover, the number of patients with dyslipidemia doubled, and a higher body mass index was noted. (4) Conclusions: Our findings suggest that patients affected by COVID-19 are at an increased risk of developing hypertension and related disorders.

## 1. Introduction

The COVID-19 pandemic continues to be a significant global health issue, even after more than three years since the first cases were reported. While most patients recover from mild to moderate respiratory illness without needing specialized treatment, it is essential to note that older individuals and those with underlying conditions like cardiovascular diseases, diabetes, chronic respiratory diseases, or neoplasia are at a higher risk of experiencing complications [1].

Patients with underlying cardiovascular disease are particularly vulnerable to the effects of COVID-19. Pre-existing cardiovascular conditions in COVID-19 patients are associated with a higher risk of mortality. Additionally, COVID-19 itself can lead to various cardiovascular complications, such as myocardial injury, arrhythmias, acute coronary syndrome (ACS), and venous thromboembolism (VTE) [2]. 

Several systematic reviews have explored the connections between cardiovascular disease (CVD) and outcomes in COVID-19 patients. Among these reviews, the largest study conducted by Luo et al. revealed that individuals with CVD had 2.65 times higher odds of mortality when infected with COVID-19 [3].

Not only patients with cardiovascular disease are affected by this infection. Studies show that COVID-19 may lead to the development of hypertension and related diseases [4,5,6,7]. Elevated blood pressure [7,8] and hypertension as post-acute sequelae of COVID-19 have been reported in several studies [9,10]. This suggests that COVID-19 could have long-term implications for cardiovascular health. However, further research is needed to fully understand the causal relationship between COVID-19 and the evolution of hypertension and related conditions [7].

The repercussions of infection with COVID-19 can be observed not only immediately, but also at a distance from the infection, months after the acute episode, leading to various symptoms. The most common ones include palpitations and chest pain. Less frequently, individuals may experience late arterial and venous thromboembolism, heart failure, strokes, or transient ischemic attacks, and myopericarditis [11]. 

The purpose of this study was to evaluate the impact of SARS-CoV-2 infection on cardiovascular diseases as well as on specific treatments.

## 2. Materials and Methods

### 2.1. Study Design and Participants

This research was conducted at the Cardiovascular Rehabilitation Clinic of the Iasi Clinical Rehabilitation Hospital (Romania), where patients with SARS-CoV-2 infection were admitted during the pandemic. Later, they were promptly taken over by our clinic in order to initiate cardio-pulmonary rehabilitation programs. From a total of 551 patients hospitalized between the 1st September and 31st December 2022, 70 patients with multiple comorbidities were selected.

The inclusion criteria were an age of over 18 and the presence of COVID infection in the last 30 days. We excluded from the research the patients who did not have a confirmed infection in their medical history, or who went through the disease more than 30 days before recruitment. We also eliminated those who had a previous admission more than three months ago, due to the fact that any acute episode could destabilize the patient’s general condition and influence the results of the analyses; those with liver failure, chronic kidney disease with a creatinine clearance less than 15 mL/min/1.73 m^2^, or cardiac pacemakers; or those who did not want to participate. The flow chart of the study group selection can be observed in Figure 1.

### 2.2. Patient Evaluation

We collected data regarding the gender, age, demographic data, BMI, comorbidities (cardiovascular, pulmonary, and metabolic conditions), and current medication (the interest was on cardiovascular antihypertensive medication and on types of drugs). The patients’ previous medical history was taken from the medical records of their prior hospitalization.

### 2.3. Ethical Approval

In order to be enrolled in the research, all of the patients had to complete a written informed consent form. Our study received approval from the Ethics Committee of the Iasi Clinical Rehabilitation Hospital (certificate of approval dated 5 May 2022).

### 2.4. Statistical Data

Data analysis was performed using SPSS 20.0 (Statistical Package for the Social Sciences, Chicago, IL, USA). The normality of the distribution was assessed for continuous variables by using the Shapiro–Wilk test. The continuous variables that had a normal distribution were presented as a mean ± standard deviation (SD), or as a number of cases (n) with a percent frequency (%) for categorical variables. The independent samples *t*-test was used when two continuous normally distributed samples were compared, while the one-way ANOVA test was applied when comparing more than two samples. When comparing categorical variables, the Fisher’s exact test or chi-square test was used in cases where the expected values of any of the contingency table cells were below 5. A two-sided *p*-value < 0.05 was considered significant for all statistical analyses. 

## 3. Results

The studied group included 70 patients, mostly women (62.9%), with an average age of 60.84 ± 12.32. The mean body mass index was 29.39 ± 5.05, and 20% of the patients were smokers (Table 1).

All cardiovascular comorbidities were considered. The most significant differences were observed in the cases of chronic kidney disease, dyslipidemia, and obesity status. Additionally, there was a statistically significant increase in the number of patients diagnosed with hypertension or chronic venous insufficiency after COVID-19 infection (Table 2). 

The data obtained led us to detail the results according to the form of infection (Table 3).

In the case of the patients with a mild form, a statistically significant decrease in renal function was observed, as well as an increase in body mass index. The number of patients with dyslipidemia doubled, but the data were not statistically significant.

The patients with a medium form of infection showed the most differences: an increase in the degree of arterial hypertension, the presence of mild tricuspid insufficiency, and an increase in body mass index. As in the case of the previous group, the most significant differences were observed in the case of renal function.

Severe infection was statistically associated with an increased body mass index and decreased kidney function.

We observed statically significant differences between age and the degree of hypertension. Moreover, even if it was without statistical significance, we noted that BMI increased in those with second- and third-degree hypertension (Table 4).

Using the Games–Howell post hoc test, we evaluated the differences between age and hypertension. Statistically significant differences were observed between those without hypertension and those with second-degree hypertension (*p* = 0.013), and, respectively, those with third-degree hypertension (*p* = 0.003). Regarding the BMI, using the same test, we observed statistically significant differences between those without hypertension and the patients diagnosed with third-degree hypertension (*p* = 0.018).

Analyzing the patients’ medication before and after the infectious episode (Table 5), statistically significant differences were obtained only in the administration of ACE inhibitors in people who went through the medium form of the disease. Even so, several aspects are worth mentioning:-Patients with a mild form required more drugs to control cardiovascular symptoms.-Those with a medium form were the category where beta-blockers and calcium channel blockers were most supplemented.-In the case of those with a severe form, a slight increase was observed in those who required beta-blocker medication and a decrease in the use of sartans.

## 4. Discussion

Ever since the identification of the COVID-19 outbreak, research efforts have been focused on evaluating optimal therapeutic interventions to attenuate COVID-19-related mortality. Similarly, studies have aimed to identify predictive factors that independently correlate with mortality in patients diagnosed with COVID-19 infection [12]. Nevertheless, information regarding the post-acute and enduring sequelae after the acute phase of COVID-19 remains limited [13]. Considering this, the present study was designed to assess both the influence of COVID-19 on cardiovascular diseases and the efficacy of specific treatments to mitigate morbidity and mortality. 

Within the COVID-19 pandemic era, researchers have focused more on hypertension and its associated pathologies [7]. For instance, in their comprehensive review, Shibata et al. investigated the relationship between COVID-19 and hypertension, elucidating that complications of this infection can be recognized as vascular disorders [14]. Furthermore, the spectrum of hypertensive disorders, encompassing cardiovascular disease and chronic kidney disease, have been identified as significant risk factors, predisposing individuals to more severe COVID-19-associated outcomes [4,5]. Over time, multiple researchers have evaluated the potential risk of hypertension after COVID-19. Notably, Zuin et al., through their meta-analysis, highlighted that the patients recovering after COVID-19 exhibited an increased risk of developing new-onset hypertension, as opposed to individuals who were not diagnosed with this infection [15]. However, compared to prior studies based on the general population, wherein the incidence ranges between 15.3 and 47.3 [16,17], this meta-analysis demonstrated a lower rate of new-onset hypertension [15]. Another study that aimed to investigate the frequency of new or worsened existing hypertension, conducted on a cohort of 200 patients, indicated that hypertension is a relatively common occurrence, with an incidence rate of approximately 16%. Additionally, the same study revealed that the patients affected by this disease had a significant risk of developing new-onset hypertension or experiencing a worsening of their existing hypertension condition. In fact, the research estimated that approximately one in every seven people affected by COVID-19 is at risk of such complications [18].

In accordance with previous studies, our analysis demonstrated an increased rate of hypertension after COVID (*p* < 0.001). Moreover, the risk was influenced by female gender (*p* = 0.050), older age, and a higher BMI.

Currently, there is still no consensus on the underlying mechanisms contributing to the emergence of new-onset hypertension following COVID-19. The existing literature data consider dysregulation within the renin–angiotensin–aldosterone system, viral persistence, pre-existing autoimmune conditions, the delayed resolution of inflammation, or residual organ damage as key mechanisms [15,19]. 

In our research, we made additional observations regarding the medication requirements of the patients. Specifically, we found that the patients with medium and severe forms of COVID-19 required an increased use of beta-blocker medication. This finding can be attributed to the mechanisms of action of these drugs and their role in the pathology of COVID-19 infection. For instance, researchers report that beta-blockers have an anti-inflammatory effect by inhibiting the release of pro-inflammatory cytokines, which helps to diminish the progression of cytokine storms in patients with severe forms of the disease. Furthermore, beta-blockers have been found to inhibit the release of catecholamines caused by the infection, thereby diminishing the development of sympathetic storms. Both cytokine storms and sympathetic storms can lead to complications in patients with severe forms of COVID-19, such as acute lung injury, acute respiratory distress syndrome, and multiple organ failure. Thus, the use of this medication, by interrupting this interaction, has the potential to reduce mortality caused by this pathology [20]. According to Heriansyah T et al., beta-blockers have been identified as beneficial due to their ability to modulate the function of juxtaglomerular cells in the kidneys, thereby blocking the entry gate of the virus. This regulation leads to a decrease in the activity of the renin–angiotensin–aldosterone system pathway and a subsequent reduction in ACE2 levels [21].

In the patients diagnosed with a moderate form of COVID-19, an increased utilization of calcium channel blockers (CCBs) was observed. The role of this medication in reducing morbidity and mortality in patients diagnosed with COVID-19 was demonstrated in a series of studies. A study conducted on 77 hospitalized COVID-19 patients showed that antihypertensive treatment with amlodipine or nifedipine was associated with reduced intubation or mechanical ventilation risk. Moreover, the patients receiving treatment with CCBs had a lower mortality risk than the control group [22]. A meta-analysis further supported these findings, which demonstrated that using CCBs is associated with a reduced mortality rate in hypertensive patients diagnosed with COVID-19 [23]. These outcomes can be attributed to the mechanism of action of CCBs. Studies demonstrated that CCBs can inhibit viral entry without affecting ACE2 expression or activity [24]. This effect is achieved by blocking the calcium ions required for fusion in cell membranes mediated by the spike protein [23]. Additionally, CCBs have a pharmacological effect on the relaxation of the pulmonary smooth muscle, thereby producing pulmonary vasodilatation. As a result, this medication helps alleviate hypoxic conditions in patients with COVID-19 [22]. 

According to statistical data, the use of ACE inhibitors (ACEis) or sartans is one of the most used therapeutic approaches for patients with hypertension. However, at the onset of the COVID-19 pandemic, there were controversies surrounding the use of these medication classes in hypertensive patients with COVID-19 [25]. Due to the interaction of ACEis or ARBs with the bradykinin pathway, concerns were raised regarding the potential increased mortality risk in hypertensive COVID-19 patients. Nevertheless, a meta-analysis conducted by Gnanenthiran et al., including 14 randomized control trials, suggested that initiating or continuing treatment with ACEis or ARBs is safe in COVID-19 hypertensive patients. Furthermore, the study revealed a lower risk of myocardial infarction among patients treated with inhibitors of the renin–angiotensin system [26]. Similarly, another meta-analysis involving 2823 patients with COVID-19 aimed to compare the effects of continuing or interrupting treatment with ACEIs/ARBs. The study observed a reduction of 41.1% in all-cause hospital mortality in patients who continued the treatment with this class of medication. The same study indicated that ACEis or ARRBs have multiple roles in inhibiting inflammation induced by the renin–angiotensin–aldosterone system axis, contributing to their beneficial effects on increasing survival rates in COVID-19 patients [27]. 

According to the World Health Organization, obesity has become a global health problem, with the number of affected individuals tripling between 1975 and 2016. In 2016, approximately 40% of the worldwide population was classified as overweight or obese [28]. The COVID-19 pandemic has presented a significant challenge, as individuals who are overweight or obese are at a higher risk of developing severe COVID-19 [29]. In the US, obesity alone accounts for 20% of COVID-19 hospitalizations, while the combination of obesity with type 2 diabetes and hypertension contributes to 60% of all COVID-19 hospitalizations [30]. In our study, we observed an increase in BMI after COVID-19 infection across all forms of the disease. This rise in obesity can be attributed to significant changes in daily routines, such as reduced physical activity and negative alterations in eating habits [29]. A systematic review involving 5,681,813 participants identified a few more risk factors for an increased BMI, including sedentarism, depression, anxiety, unhealthy lifestyle behaviors, excessive stress, sex, and being from an ethnic minority [31]. 

In line with the prevailing trend of obesity, our study demonstrated a twofold increase in the number of patients with dyslipidemia. The results align with the existing literature regarding the risk of these patients. For instance, Evan Xu et al. conducted a study to evaluate the risk of dyslipidemia in patients following COVID-19 infection. The researchers included three cohorts: 51,919 patients diagnosed with COVID-19 who survived the initial 30 days of infection, 2647 patients with no history of infection, and a historical contemporary control group consisting of 22,539,941 patients. None of the patients in these cohorts had a prior diagnosis of dyslipidemia. The study demonstrated that the patients diagnosed with COVID-19 had increased risks and burdens of incident dyslipidemia [32]. The potential mechanism underlying this association may involve the immune and inflammatory responses triggered by the infection, which could impact hepatic lipoprotein metabolism. Furthermore, changes in the oral and gut microbiomes, as well as proteomic and metabolomic profiles in patients with COVID-19, may contribute to these effects [33,34,35,36,37]. Given that COVID-19 patients experienced an increase in BMI due to changes in diet, exercise, anxiety, and depression, there may be a correlation between these two conditions [32]. However, further studies are needed to determine whether obesity may mediate the incidence of dyslipidemia, or the opposite. 

Numerous studies have indicated that COVID-19 patients are at an increased risk of experiencing a decline in kidney function following the acute phase of the infection [38]. Our study yielded similar findings, highlighting the importance of further research to elucidate the underlying mechanism driving this effect and to identify potential therapeutic interventions aimed to slow the progression of CKD.

The high prevalence of patients diagnosed with COVID-19 who require intense care measures has emphasized the need for better preventive measures to reduce the risk of complications associated with this virus, especially among patients with underlying comorbidities. Therefore, a series of vaccines became globally available, but there was a lot of debate regarding their safety and effectiveness. In Romania, patients were encouraged to receive COVID-19 vaccination, although the decision was voluntary, affording patients the autonomy to make choices regarding vaccination and their preferred vaccine. The decision was influenced by several factors, including the limited information currently available on the vaccine’s effectiveness and its potential impacts in the short-, medium-, and long-term. Due to confidentiality and privacy policies, our study did not include information on whether the patients were vaccinated against COVID-19 or not. The patients’ responses to this question were often vague, and they were evasive in providing details about their vaccination status, the type of vaccine they received, or how many doses they had. In a meta-analysis conducted by Xu et al., which analyzed 22 randomized control trials, it was demonstrated that vaccination is effective both in preventing COVID-19 infection and in reducing the risk of COVID-19 morbidity and mortality in elderly people. Moreover, the study revealed that those who received both vaccine doses had a lower risk of developing the disease. Although there were some adverse reactions to vaccination, the incidence was low, highlighting the benefits of vaccines in preventing the development of severe illness and reducing the mortality risk associated with COVID-19 [39]. 

In accordance with WHO data, 66.1% of the worldwide population has been administered the last dose of the primary vaccination series, while roughly 32% have received at least one booster or supplementary dose [40]. Thus, with the increasing number of individuals receiving COVID-19 vaccinations, a range of complications associated with the vaccine have been documented. Among these complications, cardiovascular issues have been identified as a significant concern. Notably, there has been growing attention to the potential of COVID-19 vaccination to induce an increased risk of developing hypertension [41]. 

To explore the prevalence and mechanisms of COVID-19 vaccination-induced blood pressure elevation, Angeli et al. conducted a meta-analysis that analyzed six studies, resulting in a sample size of 357,387 subjects. The authors identified 13,444 events of abnormal or elevated blood pressure among the vaccinated individuals. Furthermore, the researchers revealed a wide range of hypertensive risks associated with COVID-19 vaccination, with estimates ranging from 0.93% to 23.72%. The pooled point estimate was calculated to be 3.91%. Additionally, the study revealed that 0.6% of the vaccinated population experienced stage III hypertension, hypertensive urgencies, and hypertension emergencies [42]. 

The relationship between COVID-19 vaccines and the increased risk of developing hypertension remains uncertain and necessitates further investigation to understand the underlying mechanisms [41]. Some authors suggested that the potential link may involve the dysfunction of the counter-regulatory renin–angiotensin system axis. Moreover, studies have indicated that the increase in blood pressure after COVID-19 vaccination, mediated by the interaction between the spike protein and the angiotensin-converting enzyme 2, may be more frequent in younger subjects [43]. However, it is important to note that not all studies support this explanation, due to variability in age ranges [44,45,46]. Another hypothesis suggests that excipients present in the vaccine (e.g., polyethylene glycol) could play a role, although their concentration is considered insignificant [41,47]. The lockdown period during the COVID-19 pandemic should also be considered to contribute to the risk of developing hypertension. In this period, there was an increase in risk factors for hypertension, such as a sedentary lifestyle, unhealthy diet, anxiety, depression, and increased body mass index. 

## 5. Limitations of the Study

The strong point of this study is the fact that we evaluated the patients’ short-term outcomes after the infections, many of them being lost from the record in other centers where cardio-respiratory recovery is not performed.

The present research was performed on a small group of patients. We still do not have data related to the impact of the infection on cardiovascular diseases, mainly on arterial hypertension, so we can comment only on the changes that occurred one month after the acute episode.

## 6. Conclusions

Over the last few years, infection with COVID-19 has become a widely debated topic in the medical literature due to its consequences across various systems. 

The effects of the infection observed in this study were an increase in the degree of hypertension and the need to change treatment by choosing medication to control the heart rate (beta-blockers and calcium blockers). One month after the infection, a higher BMI, deterioration of renal function, and alteration of the lipid profile were observed.

Our conclusions lead to the idea of the infection amplifying cardiovascular risk factors, in addition to the direct effect that the infection has on blood pressure.

The implication of this increased risk on the global burden of cardiovascular diseases, as well as its impact on the healthcare system and costs, will require further investigation and the need for the longer-term follow-up of patients with comorbidities and added risk factors.

## Figures and Tables

**Figure 1 jcm-12-06538-f001:**
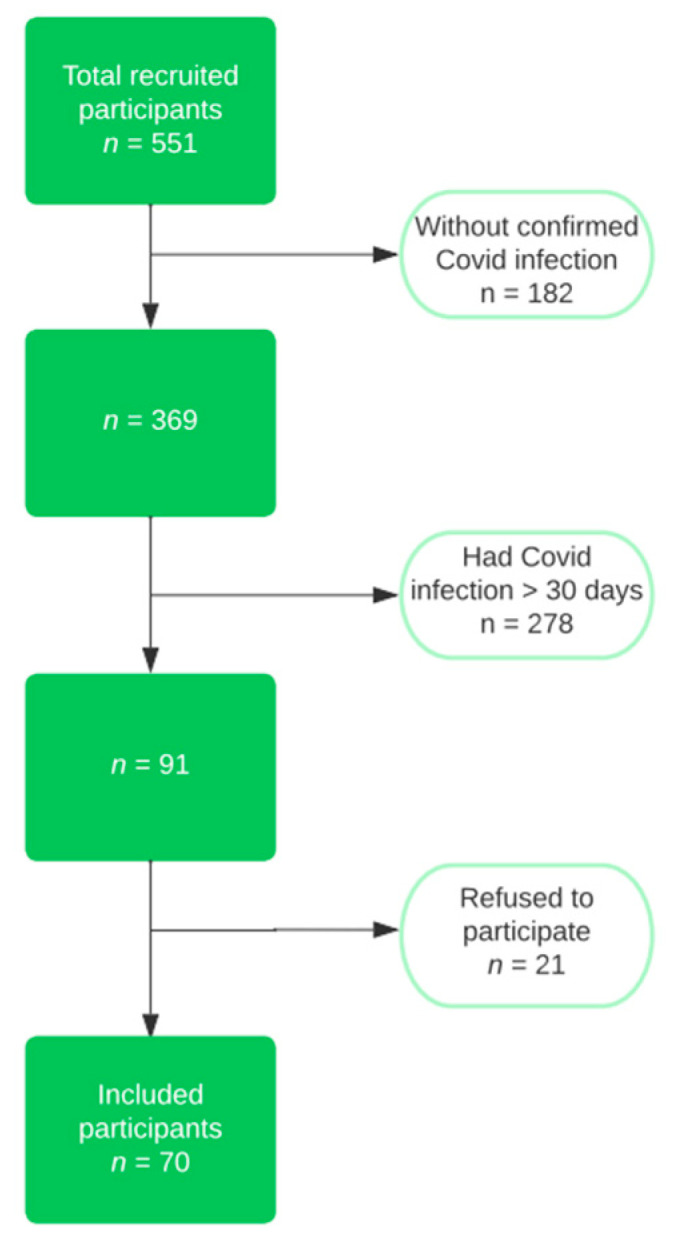
Flow chart of the study group selection.

**Table 1 jcm-12-06538-t001:** General characteristics according to the form of the disease in the study group.

General Characteristics		Total (n = 70)	Mild (n = 19)	Medium (n = 39)	Severe (n = 12)	*p*-Value
Age, years (mean ± SD)		60.84 ± 12.32	62.52 ± 12.29	59.69 ± 10.95	61.91 ± 16.72	0.681
Gender, n (%)						0.866
	Male	26 (37.1)	8 (42.1)	14 (35.9)	4 (33.3)	
	Female	44 (62.9)	11 (57.9)	25 (64.1)	8 (66.7)	
BMI, kg/m^2^ (mean ± SD)		29.39 ± 5.05	29.35 ± 3.79	29.35 ± 5.68	29.59 ± 4.99	0.989
Smoking		14 (20%)	4 (21.1)	8 (20.5)	2 (16.7)	0.791
Place of origin						
	Rural	27 (38.6)	6 (31.6)	18 (46.2)	3 (25.0)	0.912
	Urban	43 (61.4)	13 (68.4)	21 (53.8)	9 (75.0)	

BMI—body mass index.

**Table 2 jcm-12-06538-t002:** Comorbidities in the studied group, before and after infection.

Comorbidities	Pre-COVID (n = 70)	Post-COVID (n = 70)	*p*-Value
Hypertension, n (%)	48 (68.57)	63 (90)	0.005
Chronic ischemic heart disease, n (%)	10 (14.28)	17 (24.3)	0.198
Tricuspid regurgitation, n (%)	3 (4.28)	9 (12.85)	0.128
Mitral regurgitation, n (%)	7 (10)	18 (25.71)	0.037
Chronic cardiac failure, n (%)	10 (14.28)	15 (21.42)	0.378
By-pass, n (%)	5 (7.14)	5 (7.14)	1.000
Atrial fibrillation, n (%)	6 (8.57)	5 (7.1)	1.000
Transient ischemic attack, n (%)	7 (10)	7 (10)	1.000
Arteriosclerosis obliterans, n (%)	2 (2.85)	8 (11.42)	0.166
Chronic venous insufficiency, n (%)	9 (12.85)	21 (30)	0.022
Dyslipidemia, n (%)	16 (22.85)	33 (47.14)	0.004
Obesity, n (%)	13 (18.57)	46 (65.71)	<0.001
Diabetes mellitus, n (%)	13 (18.57)	13 (18.57)	1.000
Asthma, n (%)	2 (2.85)	2 (2.85)	1.000
Chronic obstructive pulmonary disease, n (%)	4 (5.51)	4 (5.51)	1.000
Chronic kidney disease, n (%)	4 (5.71)	69 (98.57)	<0.001

**Table 3 jcm-12-06538-t003:** Comorbidities in the studied group, depending on the form of the disease.

		Mild			Medium			Severe		
		Pre- COVID	Post- COVID	*p*-Value	Pre- COVID	Post- COVID	*p*-Value	Pre- COVID	Post- COVID	*p*-Value
HBP, n (%)				0.138			0.033			0.590
	1	3 (15.8)	4 (21.1)		3 (7.7)	8 (20.5)		2 (16.7)	3 (25)	
	2	6 (31.6)	5 (26.3)		13 (33.3)	14 (35.9)		2 (16.7)	3 (25)	
	3	5 (26.3)	9 (47.4)		9 (23.1)	12 (30.8)		5 (41.7)	5 (41.7)	
CHD, n (%)		2 (10.5)	6 (31.6)	0.232	8 (20.5)	10 (25.6)	0.789	0 (0)	1 (8.3)	1.000
TR, n (%)				1.000			0.021			1.000
	Mild	1 (5.3)	1 (5.3)		0 (0)	4 (10.3)		1 (8.3)	2 (16.7)	
	Moderate	0 (0)	0 (0)		0 (0)	1 (2.6)		1 (8.3)	1 (8.3)	
MR, n (%)				0.660			0.138			0.246
	Mild	2 (10.5)	4 (21.1)		4 (10.3)	10 (25.6)		1 (8.3)	2 (16.7)	
	Moderate	0 (0)	0 (0)		0 (0)	0 (0)		0 (0)	2 (16.7)	
CHF, n (%)				0.660			0.347			1.000
	NYHA I	0 (0)	0 (0)		0 (0)	0 (0)		1 (8.3)	1 (8.3)	
	NYHA II	1 (5.3)	3 (15.8)		3 (7.7)	5 (12.8)		2 (16.7)	1 (8.3)	
	NYHA III	1 (5.3)	1 (5.3)		0 (0)	2 (5.1)		1 (8.3)	1 (8.3)	
	NYHA IV	0 (0)	0 (0)		1 (2.6)	0 (0)		0 (0)	0 (0)	
By-pass, n (%)				1.000			1.000			-
	1	0 (0)	0 (0)		2 (5.1)	1 (2.6)		0 (0)	0 (0)	
	3	1 (5.3)	1 (5.3)		2 (5.1)	2 (5.2)		0 (0)	0 (0)	
	4	0 (0)	0 (0)		0 (0)	1 (2.6)		0 (0)	0 (0)	
AF, n (%)		1 (5.3)	0 (0)	-	3 (7.7)	4 (10.3)	1.000	2 (16.7)	1 (8.3)	1.000
TIA, n (%)		1 (5.3)	1 (5.3)	1.000	3 (7.7)	3 (7.7)	1.000	3 (25)	3 (25)	1.000
AO, n (%)				-			0.313			0.478
	I	0 (0)	0 (0)		0 (0)	3 (7.7)		0 (0)	0 (0)	
	II	0 (0)	0 (0)		1 (2.6)	2 (5.1)		0 (0)	2 (16.7)	
	IV	0 (0)	0 (0)		1 (2.6)	1 (2.6)		0 (0)	0 (0)	
CVI, n (%)				0.447			0.160			-
	CEAP 1	0 (0)	0 (0)		0 (0)	1 (1)		1 (8.3)	0 (0)	
	CEAP 2	2 (10.5)	3 (15.8)		1 (2.6)	3 (7.7)		0 (0)	1 (8.3)	
	CEAP 3	1 (5.3)	2 (10.5)		0 (0)	4 (10.3)		0 (0)	3 (25)	
	CEAP 4	0 (0)	1 (5.3)		3 (7.7)	3 (7.7)		0 (0)	0 (0)	
	CEAP 6	0 (0)	0 (0)		1 (2.6)	0 (0)		0 (0)	0 (0)	
HC, n (%)		5 (26.3)	10 (52.6)	0.184	7 (17.9)	17 (43.6)	0.026	4 (33.3)	6 (50)	0.680
BMI, n (%)				0.012			<0.001			0.012
	Overweight	2 (10.5)	6 (31.6)		1 (2.6)	7 (17.9)		0 (0)	3 (25)	
	I	2 (10.5)	7 (36.8)		4 (10.3)	11 (28.2)		0 (0)	3 (25)	
	II	0 (0)	1 (5.3)		1 (2.6)	4 (10.3)		0 (0)	1 (8.3)	
	III	1 (5.3)	0 (0)		1 (2.6)	2 (5.1)		1 (8.3)	1 (8.3)	
DM, n (%)		2 (10.5)	2 (10.5)	1.000	7 (17.9)	8 (20.5)	1.000	4 (33.3)	4 (33.3)	1.000
Asthma, n (%)		1 (5.3)	1 (5.3)	1.000	1 (2.6)	1 (2.6)	1.000	12 (100)	12 (100)	1.000
COPD, n (%)		1 (5.3)	2 (10.5)	1.000	2 (5.1)	2 (5.1)	1.000	1 (8.3)	1 (8.3)	1.000
CKD, n (%)				<0.001			<0.001			<0.001
	1	0 (0)	2 (10.5)		0 (0)	1 (2.6)		0 (0)	2 (17.7)	
	2	0 (0)	11 (57.9)		0 (0)	27 (69.2)		0 (0)	6 (50)	
	3	0 (0)	6 (31.6)		2 (5.1)	10 (25.6)		1 (8.3)	3 (25)	
	4	1 (5.3)	0 (0)		0 (0)	0 (0)		0 (0)	1 (8.3)	

HBP—high blood pressure; CHD—coronary heart disease; TR—tricuspid regurgitation; MR—mitral regurgitation; CF—cardiac failure; AF—atrial fibrillation; TIA—transient ischemic attack; AO—arteriosclerosis obliterans; CVI—chronic venous insufficiency; HC—hypercholesterolemia; BMI—body mass index; DM—diabetes mellitus; COPD—chronic obstructive pulmonary disease; and CKD—chronic kidney disease.

**Table 4 jcm-12-06538-t004:** Risk factors associated with hypertension.

	Without	First-Degree	Second-Degree	Third-Degree	
Age	53.42 ± 5.79	50.00 ± 10.70	64.72 ± 11.69	65.80 ± 10.35	<0.001
BMI	25.80 ± 2.74	27.97 ± 4.34	30.05 ± 5.26	30.63 ± 5.29	0.080

**Table 5 jcm-12-06538-t005:** Cardiovascular medication before and after the infection.

Medication	Mild			Medium			Severe		
	Pre- COVID	Post- COVID	*p*-Value	Pre- COVID	Post- COVID	*p*-Value	Pre- COVID	Post- COVID	*p*-Value
ACE inhibitors, n (%)	3 (15.8)	5 (26.3)	0.693	6 (15.4)	1 (2.6)	0.048	2 (16.7)	2 (16.7)	1.000
Sartans, n (%)	2 (10.5)	3 (15.8)	1.000	5 (12.8)	8 (20.5)	0.545	6 (50)	3 (25)	0.400
Beta-blockers, n (%)	7 (36.8)	8 (42.1)	1.000	16 (41)	24 (61.5)	0.112	6 (50)	8 (66.7)	0.680
Calcium channel blockers, n (%)	7 (36.8)	9 (47.4)	0.743	6 (15.4)	13 (33.3)	0.112	5 (41.7)	6 (50)	1.000
Diuretic, n (%)	5 (26.3)	4 (21.1)	0.157	10 (25.7)	6 (15.4)	0.516	5 (41.7)	5 (41.7)	1.000
Central blockers, n (%)	1 (5.3)	1 (5.3)	1.000	1 (2.6)	2 (5.1)	1.000	1 (8.3)	1 (8.3)	1.000

## Data Availability

The data presented in this study are available upon request from the corresponding author.

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
