# Peer review of "Evolution of Cardiovascular Risk Factors in Post-COVID Patients"

_jcm, 2023, doi:10.3390/jcm12206538_

Round 1
Reviewer 1 Report
1. Introduction can be reformed and discuss more about how CVD risk factors and COVID-19 interact
2. Clear need of the study is required for e.g. are the authors only including long COVID or are they looking into any COVID patient withe the disease
3. Inclusion and exclusion criteria needs to be clear
4. Table 3 I see people with Co-Morb , but the participants number are really low (NYHA) ? why ? applies to most of the variables in the table
5. Use a Graphical Abstract for the study
6. Under discussion the authors have discussed more about Drug and its interaction , what about other interaction effect , vaccine status among the participants and its interaction ?
7. Accordingly conclusion needs to be reframed
English needs drafting pls upload a certificate after the correction done by an professional
Author Response
Dear reviewer,
We shaped the article in accordance with the requirements of all reviewers. Thank you for your suggestions, as well as for the expertise, effort, and time spent.
Point 1. Introduction can be reformed and discuss more about how CVD risk factors and COVID-19 interact
Response 1: We added informations about the CVD risk factors and Covid-19. Thank you for your suggestion!
Point 2. Clear need of the study is required for e.g. are the authors only including long COVID or are they looking into any COVID patient withe the disease
Response 2: The study included patients who were diagnosed with Sars-Cov2 infection in the last 30 days. We could not take into account the classification of long-Covid, considering that this is given when the symptoms persist for more than 12 weeks after the infectious episode.
Thank you for your question!
Point 3. Inclusion and exclusion criteria needs to be clear
Response 3: We have modified the paragraph so that the inclusion and exclusion criteria are more clearly stated. We hoped that now your requirement is fulfulled. Thank you!
Point 4. Table 3 I see people with Co-Morb , but the participants number are really low (NYHA) ? why ? applies to most of the variables in the table
Response 4: The Clinical Rehabilitation Hospital from Iasi is the only one in the North-Eastern region of Romania that has a cardio-vascular rehabilitation program for post-Covid patients. Even so, the degree of addressability was very low, a fact that draws attention to the need to keep records of patients after the acute episode of the disease, especially considering that post-Covid medical rehabilitation must be initiated. We wanted to present the profile of the patients who were hospitalized in the clinic during the mentioned period, and the data are presented in the related table.
Thank you for your question!
Point 5. Use a Graphical Abstract for the study
Response 5: We added a Graphical Abstract. Thank you for your suggestion!
Point 6. Under discussion the authors have discussed more about Drug and its interaction , what about other interaction effect , vaccine status among the participants and its interaction ?
Response 6: The aim of the authors was to quantify the cardiovascular disease, not other types, so that our objective was found in the data presented.
Vaccinated/non-vaccinated status could not be taken into account, considering that the data were confidential, and some patients did not want to express their answer.
Thank you for your question!
Point 7. Accordingly conclusion needs to be reframed
Response 7: We reframed the conclusions as you requested. Thank you for your suggestion!
Reviewer 2 Report
The manuscript titled “Evolution of cardiovascular risk factors in post-COVID patients” investigated the risk of patients infected with COVID-19 to develop cardiovascular diseases. There are some significant concerns: 1) COVID-19 affecting the rate of new-onset cardiovascular diseases is not novel since it has been widely discussed. 2) The participants selected were infected with COVID-19 in the last 30 days, which could not indicate patients were in the recovery phase. 3) The number of patients was too small, and the conclusion may not be convincing.
Author Response
Dear reviewer,
We shaped the article in accordance with the requirements of all reviewers. We will try in our present answer to explain the rationale of this study. Thank you for your suggestions, as well as for the expertise, effort and time spent.
Point 1. COVID-19 affecting the rate of new-onset cardiovascular diseases is not novel since it has been widely discussed.
Response 1: The present study is an epidemiological one that had as its aim both the quantification of early cardiovascular damage and the evolution of pre-existing cardiovascular diseases in patients from the North-East of Romania. Considering that in this region, the number of studies of this kind is low, we thought such a work appropriate.
Thank you!
Point 2. The participants selected were infected with COVID-19 in the last 30 days, which could not indicate patients were in the recovery phase.
Response 2: The purpose of the research was to make an early assessment, to observe the immediate impact of the infection with Sars-Cov2, following that these patients, depending on the subsequent assessments, should be included in the cardio-vascular rehabilitation program.
Thank you for your suggestion!
Point 3. The number of patients was too small, and the conclusion may not be convincing.
Response 3: Another aspect that we considered extremely important was precisely the small number of patients that we could introduce into the study. The Clinical Rehabilitation Hospital from Iasi is the only one in the North-Eastern region of Romania that has a cardio-vascular rehabilitation program for post-Covid patients. Even so, the degree of addressability was very low, a fact that draws attention to the need to keep records of patients after the acute episode of the disease, especially considering that post-Covid medical rehabilitation must be initiated. In perspective, the patients will be followed up after a longer period of time, being the object of study of some subsequent researches.
Thank you!
Hope we have touched all the points you asked us to change.
If there are any other changes you consider we should make, please let us know.
Yours sincerely,
All the authors
Round 2
Reviewer 1 Report
Kindly reform the introduction part , most of the comments only addressed partly
the graphical abstract is also not visible ?
Inclusion /exclusion not mentioned clearly
Discussion and conclusion not reframed accordingly
Limitation to the study ?
Minor revision
Author Response
Dear reviewer,
Thank you for all your comments.
- We have modified the introduction part. Thank you!
- We have added more details and explained better the inclusion and exclusion criteria.
- We have added in the Discussion section more details regarding the vaccination. Thank you!
- We have added a Study Limitation section.
Hope we have touched all the points you asked us to change.
If there are any other changes you consider we should make, please let us know.
Yours sincerely,
All the authors
Reviewer 2 Report
No further comments!
Author Response
Dear reviewer,
Thank you for all your comments.
If there are any other changes you consider we should make, please let us know.
Yours sincerely,
All the authors